# Antifungal Activity of Eugenol Derivatives against *Botrytis Cinerea*

**DOI:** 10.3390/molecules24071239

**Published:** 2019-03-29

**Authors:** Andrés F. Olea, Angelica Bravo, Rolando Martínez, Mario Thomas, Claudia Sedan, Luis Espinoza, Elisabeth Zambrano, Denisse Carvajal, Evelyn Silva-Moreno, Héctor Carrasco

**Affiliations:** 1Instituto de Ciencias Químicas Aplicadas, Facultad de Ingeniería, Universidad Autónoma de Chile, San Miguel, Santiago 8900000, Chile; andres.olea@uautonoma.cl; 2Departamento de Ciencias Químicas, Universidad Andrés Bello, Viña del Mar 2520000, Chile; angebravor@gmail.com (A.B.); rmartinez@unab.cl (R.M.); m.thomaselgueda@gmail.com (M.T.); claudiasedan@hotmail.com (C.S.); 3Departamento de Química, Universidad Técnica Federico Santa María, Valparaíso 2340000, Chile; Luis.espinozac@usm.cl; 4Instituto de Ciencias Biomédicas, Facultad de Ciencias de la Salud, Universidad Autónoma de Chile, San Miguel, Santiago 8900000, Chile; e.zambrano01@ufromail.cl (E.Z.); denisse.carvajals@gmail.com (D.C.)

**Keywords:** *Botrytis cinerea*, eugenol, mycelial growth, resistant isolate, reactive oxygen species, chemical fungicides

## Abstract

*Botrytis cinerea* is a worldwide spread fungus that causes the grey mold disease, which is considered the most important factor in postharvest losses in fresh fruit crops. Consequently, the control of gray mold is a matter of current and relevant interest for agricultural industries. In this work, a series of phenylpropanoids derived from eugenol were synthesized and characterized. Their effects on the mycelial growth of a virulent and multi-resistant isolate of *B. cinerea* (PN2) have been evaluated and IC_50_ values for the most active compounds range between 31–95 ppm. The antifungal activity exhibited by these compounds is strongly related to their chemical structure, i.e., increasing activity has been obtained by isomerization of the double bond or introduction of a nitro group on the aromatic ring. Based on the relationship between the fungicide activities and chemical structure, a mechanism of action is proposed. Finally, the activity of these compounds is higher than that reported for the commercial fungicide BC-1000 that is currently employed to combat this disease. Thus, our results suggest that these compounds are potential candidates to be used in the design of new and effective control with inspired natural compounds of this pathogen.

## 1. Introduction

*B. cinerea* is a serious worldwide problem because it causes high losses in pre- and postharvest fresh fruit crops. The high economic losses associated with *Botrytis* infection represents a growing burden for the agricultural industry [1]. To reduce this effect, a series of control mechanisms have been developed, the application of chemical fungicides being the most spread and more used [2]. Currently, the fungicides available to control *B. cinerea* on grapevines are hydroxianilides (fenhexamid), anilinopyrimidines (cyprodinil and pyrimethanil), dicarboximide (iprodione), carboxamides (boscalid), strobilurin, phenylpirroles (fludioxonil) and some inhibitors of ergosterol synthesis [3]. However, despite the variety of action mechanisms of these chemical fungicides their indiscriminate use has led to the appearance of resistant isolates. This undesirable outcome has prompted the search of new and effective fungicide agents, mainly from natural resources [4,5,6,7].

Eugenol (4-allyl-2-methoxyphenol) (**1**) is a major component of essential oil isolated from the *Eugenia caryoplyllata*. Much attention has been dedicated to this molecule because it exhibits many different biological activities, such as analgesic [8,9], antimicrobial [10,11], antifungal [12,13], antioxidant [14], anti-inflammatory [15], anti-carcinogenic [16], anti-mutagenic and even repellent properties [17,18].

Various studies of the antifungal activity of eugenol and some analogues (Figure 1) against phytopathogenic and human fungi, have been reported [19,20,21,22,23,24,25].

Zemek et al. [19] reported that **1** is almost inactive against *Saccharomyces cerevisiae*, *Candida albicans* and *Aspergillus niger* (MICs around 3000 ppm), while **2** exhibits a moderate inhibitory effect on the same fungi with MICs ranging between 100 and 250 ppm in both dilution methods. On the other hand, Kubo et al. [20] reported that **1** possess moderate activity against *S. cerevisiae*, *Candida subtilis*, *Pityrosporum ovale*, and *Penicillum chrysogenum*, with MICs between 100 to 800 ppm in both dilution methods with shaking, *P. ovale* being the most sensitive fungus; **1** and **3** exhibit moderate activity against *C. albicans*, i.e., measured MICs were 800 and 200 ppm, respectively [21]; **3** is active against *S. cerevisiae* at 200 ppm [22]. Eugenol has also shown activity against *Alternaria sp.* and *P. chrysogenum*, but is inactive against *A. niger*, *Botryosphaeria rhodina* or *Rhizoctonia* sp. in agar diffusion assays [23].

Recently, it has been found that **1** exhibits effective antifungal activity against *Aspergillus*, *Penicillium*, *Emericella* and *Fusarium* spp., at concentrations between 100 to 150 ppm. This activity has been attributed mainly to the phenolic group [24].

In previous work, we have assessed the antifungal properties of a series of phenylpropanoids including eugenol, safrole and some synthetic derivatives against a panel of opportunistic pathogenic fungi humans. One of these derivatives, 4-allyl-2-methoxy-5-nitrophenol, possesses a high activity on *C. albicans* and non-albicans *Candida* spp., *Cryptococcus neoformans* and *dermatophytes*. Additionally, it was shown that this molecule did not bind to the main sterol of the fungal membrane up to 250 ppm [25].

However, the antifungal activity of eugenol against *B. cinerea* has been barely studied. The effect of eugenol and its essential oils against four apple pathogens, including *B. cinerea*, have been evaluated in vitro and in vivo [26]. The MIC value determined for eugenol, incorporated in malt extract agar medium, was 2000 ppm, whereas in the saturated atmosphere of volatile eugenol a complete inhibition of mycelial growth of four pathogens was obtained at 150 µL of volatile eugenol per liter of air volume. No effect on the germination of all tested pathogens at room temperature was found for eugenol [26].

Recently, the in vitro activity of eugenol on *B. cinerea* has been studied. The IC_50_ value measured on mycelial radial growth of *B. cinerea* was 38.6 ppm; and no bioactivity against conidia germination was observed. Eugenol induces the generation of H_2_O_2_ and increases a free Ca^2+^ concentration in the cytoplasm. These results strongly support the idea that the antifungal activity of eugenol is due to membrane binding and permeability alteration, leading to destabilization and disruption of the plasma membrane [27].

Herein, we report the synthesis of a series of eugenol derivatives (Figure 2), six of them new compounds (**8**–**13**), and their evaluation as inhibitors of mycelial growth of a virulent and multidrug-resistant native isolate (PN2). A relationship between the chemical structures of these compounds and their activities is discussed.

## 2. Results

A series of eugenol derivatives have been synthesized and their antifungal properties have been evaluated (see Figure 2). The synthesis and spectroscopic characterization of compounds **4**–**7**, and **14**–**17** have been previously described [25,26]. In this work, new compounds **8**–**13** have been synthesized and a full spectroscopic characterization is given. Eugenol derivatives **8**, **10** and **11** were obtained from 4-allyl-2-methoxy-6-nitrophenol (**7**) with high reaction yields (55–80%). The reduction of aromatic nitro group in **7** and **11** gives anilines **8** and **12** with 70 and 45 % of reaction yield, respectively. Finally, the acetylation of **8** and **12** leads to **9** and **13** with 70% and 53% reaction yield, respectively. All these syntheses were carried out following standard procedures.

The antifungal activity of eugenol and its derivatives was studied using PN2, a resistant *Botrytis cinerea* native isolate. This isolate was obtained from cherry fruit and presents resistance to fenhexamid, iprodione, pyrimethanil and boscalid. The virulence of this isolate was assessed using a tomato and the results are shown in Figure 3.

Results shown in Figure 3 indicate that an important injury is produced on tomato fruits after four days of inoculation with all tested isolates of *B. cinerea*. The damage is proportional to the post-inoculation time and therefore we may conclude that PN2 is a virulent and resistant native isolate.

### Evaluation of Antifungal Activity of Eugenol and Eugenol Derivatives on Botrytis cinerea

The antifungal activity of eugenol and its derivatives was evaluated in radial growth measurements in malt-yeast medium. The results shown in Figure 4 indicate that the eugenol derivatives affect mycelial growth of *B. cinerea* in a concentration-dependent manner. The percentage of growth inhibition was determined by measuring the mycelial growth in the absence and presence of different concentrations of tested compounds (see Experimental section). Results obtained for the most active compounds are shown in Figure 5.

## 3. Discussion

The antifungal activity was evaluated by the measurement growth inhibition. The fitting of percentage of inhibition to a dose-response curve allows the obtention of IC_50_ values. The calculated values of IC_50_ for all assayed compounds are given in Table 1 and range from 31 to 440 ppm.

The results indicate that the chemical modification of eugenol induces important changes on the antifungal activity of this compound. The antifungal activity of eugenol has been attributed to disruption of the fungal membrane structure, mainly by accumulation of eugenol in the phospholipid bilayer due to its lipophilic character. This interaction alters fluidity and permeability of fungal membranes and affects the function of important membrane-bound enzyme or protein [27].

In this work, the chemical modification of eugenol includes changes in the existing functional groups (Figure 2, compounds **4**–**6**) and the addition of nitrogen-containing group on the aromatic ring (Figure 2, compounds **7**–**17**). In the first group, isomerization of the double bond (compare chemical structures of compounds **1** and **4**, Figure 2) has been carried out, and there is almost no effect on eugenol lipophilicity. However, this small structural modification reduces the IC_50_ value from 149 to 109 ppm. In other words, an increase of growth inhibition is induced in eugenol derivatives by conjugation of the side chain double bond with the aromatic system. As lipophilicity is quite similar, the observed increase on the antifungal activity must be explained in terms of an additional mechanism by which molecules **4**–**6** acts on fungus. A possible explanation is the enhanced Michael-type reaction that could occur between the side propenyl chain and biological nucleophiles. The reaction mechanism would be like that proposed for menadione, a lipid-soluble molecule that causes changes in plasma membrane permeability. Briefly, menadione and compounds **4**–**6** might react with nucleophilic molecules through a Michael-type reaction generating an intermediate reactive species that later is stabilized by reaction with an acid. A comparison of this mechanism, for menadione and isoeugenol, is depicted in Scheme 1.

Further reductions are reached by changing the hydroxyl group to acetyl (**5**) or methoxy (**6**) groups, i.e., by decreasing the polarity of isoeugenol.

On the other hand, the addition of nitrogen-containing group on the aromatic ring (Figure 2, compounds **7**–**17**) produces different effects on the antifungal activity. Results shown in Table 1 indicate that introduction of nitro group decreases the IC_50_ value in a factor of 2.5–4.5 (compounds **7**, **14** and **15** in Table 1). However, this effect disappears when the nitro group is replaced by amine or acetamide groups (compare IC_50_ values of **7** and **8**, and **11** and **12**, respectively). Considering that the polarity of nitro compounds is higher than that of related eugenol derivatives, the antifungal activity of these compounds, associated to lipophilic concentration in the membrane, should be lower than that shown by eugenol. Therefore, additional mechanisms must be considered to explain the lowest values of IC_50_ obtained for these compounds. The nitro group is very strong electron withdrawing and might react directly with double bonds substituted with other electron withdrawing groups. In addition, it has been shown that aromatic nitro compounds produce reactive oxygen species (ROS) in an enzymatic-mediated process [28]. Both processes can lead to a chemical disruption of the membrane palisade. The different activities shown by the nitro eugenol derivatives might be attributed to the location of these compounds in the membrane or electronic effect affecting the efficiency of both reactions. Figure 6 depicts a schematic representation of different locations of bioactive compounds into the fungus cell wall.

The compound-position and orientation in the membrane is determined by the chemical structure of the adsorbed molecule. Eugenol and derivatives must orientate themselves with the side alkyl chain parallel to the membrane chains, and the hydroxyl group anchored to the polar surface. Consequently, the nitro group in compound **7** is located near the surface, whereas in compound **14** the nitro group is buried into the palisade. In the latter, the spatial arrangement could enhance the nucleophilic attack of double bonds on the electrophilic nitro groups, affecting the membrane permeability and probably the transport of electrons. A similar configuration should be adopted by **15**. This mechanism of action reduces IC_50_ values from 149 (eugenol) to 62, 31 and 39 ppm for **7**, **14** and **15**, respectively. The presence of other groups in the aromatic ring may affect the reactivity of nitro compounds either by modifying the electronic distribution or the location in the palisade. For example, the replacement of the hydroxyl group by acetyl decreases strongly the activity of nitroeugenol derivatives, i.e., compares IC_50_ values of **7** and **10**, **14** and **16**, **15** and **17**. These results can be attributed to the highest electronic attraction of the hydroxyl group, which enhances the nucleophilic reaction with double bonds.

The compound-position and orientation in the membrane is determined by the chemical structure of the adsorbed molecule. Eugenol and derivatives must orientate themselves with the side alkyl chain parallel to the membrane chains, and the hydroxyl group anchored to the polar surface. Consequently, the nitro group in compound **7** is located near the surface, whereas in compound **14** the nitro group is buried into the palisade. In the latter, the spatial arrangement could enhance the nucleophilic attack of double bonds on the electrophilic nitro groups, affecting the membrane permeability and probably the transport of electrons. A similar configuration should be adopted by **15**. This mechanism of action reduces IC_50_ values from 149 (eugenol) to 62, 31 and 39 ppm for **7**, **14** and **15**, respectively. The presence of other groups in the aromatic ring may affect the reactivity of nitro compounds either by modifying the electronic distribution or the location in the palisade. For example, the replacement of the hydroxyl group by acetyl decreases strongly the activity of nitroeugenol derivatives, i.e. compares IC_50_ values of **7** and **10**, **14** and **16**, **15** and **17**. These results can be attributed to the highest electronic attraction of the hydroxyl group, which enhances the nucleophilic reaction with double bonds.

On the other hand, the chemical reactivity of the nitro group decreases by substitution of the ortho hydroxy group by a methoxy group (compounds **7** and **11**) due to its lower positive inductive effect, but it takes the molecule deeper in the membrane. These two opposite factors induce a slight decrease on the IC_50_ value.

Thus, our results indicate that the growth inhibition observed for eugenol derivatives evaluated in this work could be a consequence of two different and parallel mechanisms: (i) accumulation in fungal membrane by lipophilic interaction and (ii) Michael-type reactions between eugenol derivatives and membrane components or ROS production by enzymatic reduction of nitro compounds.

The second mechanism seems to be more important and probably these chemical reactions induce the production of reactive oxygen species. It is known that ROS are generated in cells by a variety of processes associated with the normal function of cells or induced by the presence of exogenous cytotoxic substrates. In addition, plants produce high amounts of ROS to avoid fungus infection, and therefore the inability of pathogen to reduce the ROS level may be the cause of fungicide effect [29]. To elucidate this hypothesis, the effect of the most active nitroeugenol derivatives on ROS production has been assessed. Figure 7 shows the luminescence yields measured in the absence and presence of nitro compounds and menadione that is used as a positive control. Luminescence is a measure of H_2_O_2_ concentration or ROS production.

The results shown in Figure 7 indicate that nitroeugenol derivatives induce ROS production in conidia in an amount equal or larger than that generated by menadione (positive control), which is a model cytotoxic compound that induces ROS production inside the cell by mitochondria uncoupling [30]. Interestingly, the magnitude of ROS production follows the same order of IC_50_ values, i.e., 15 > 11 > 7. In other words, the formation of ROS correlates quite well with the inhibition activity suggesting that nitro compounds are acting on the fungus membrane by generating reactive oxygen species and by alteration of membrane in a similar way to that reported for eugenol both in fungi and bacteria [31].

## 4. Materials and Methods

### 4.1. Chemistry

Unless otherwise stated, all chemical reagents were purchased with the highest commercially available purity (Merck, Darmstadt, Germany or Aldrich, St. Louis, MO, USA) and were used without previous purification. Melting points were measured on a Stuart-Scientific SMP3 apparatus and are uncorrected. IR spectra were recorded as thin film or KBr pellets in a Nicolet Impact 420 spectrometer (Thermo Scientific, San Jose, CA, USA). ν_max_ values are expressed in cm^−1^. ^1^H- and ^13^C-NMR spectra were recorded in CDCl_3_ solutions and referenced to the residual peak of CHCl_3_ at δ = 7.26 ppm and δ = 77.00 ppm for ^1^H and ^13^C, respectively, on a Bruker Avance 400 Digital NMR spectrometer (Bruker, Rheinstetten, Germany), operating at 400.1 MHz for ^1^H and 100.6 MHz for ^13^C. Chemical shifts are reported in δ ppm and coupling constants (*J*) are given in Hz. Silica gel (Merck 200–300 mesh) was used for column chromatography (CC) and silica gel plates and HF-254 for thin layer chromatography (TLC). Spots were detected on TLC by heating after spraying with 25% H_2_SO_4_ in H_2_O.

### 4.2. Synthesis

Eugenol was isolated from cloves essence, according to the standard procedure [32]. Synthesis of new eugenol derivatives are shown in Scheme 2.

#### General Procedures

Reduction of nitroeugenol derivatives to anilines. Nitroeugenol derivatives **7** and **11** were transformed to the respective aniline (**8** and **12**, respectively) by the following procedure. Zn in powder (100 mg, previously treated with HCl 5%) and ammonium formate (1 mL) were added to a stirred solution of nitroeugenols (**7**, 200 mg, 9.6 × 10^−4^ mol; **11**, 100 mg, 4.48 × 10^−4^ mol) in methanol (5.0–10 mL). The solution was stirred at room temperature for 5 h, and complete disappearance of the starting product was confirmed by TLC (AcOEt:n-hexane, 1:3). The reacted mixture was filtered, and the solvent evaporated in vacuum. The crude reaction product was dissolved in AcOEt (10 mL) and the organic phase was washed with brine (3 × 10 mL), dried with anhydrous Na_2_SO_4_ and vacuum evaporated. The pure product was obtained by CC.

Acetylation reaction. Phenols **7**, **8** and aniline **12** were reacted with acetic anhydride to give **10**, **9**, and **13**, respectively, by the following synthetic procedure. To a stirred solution of phenol (100 mg, 5.6 × 10^−4^ mol) or aniline (100 mg, 4.48 × 10^−4^ mol) in dichloromethane (15 mL) was added 4-*N*,*N*-dimethylaminopyridine (DMAP) (10 mg, 8.2 × 10^−5^ mol). Acetic anhydride (0.25 mL) was added and the reaction was left to continue at room temperature (1.5–2 h for phenols; 1.5 h for aniline). After this period, complete disappearance of the starting product was confirmed by means of the TLC (ethyl acetate: n-hexane 1:3). Aqueous potassium hydrogen sulphate solution (10%, 20 mL) was added, and the organic phase was extracted with dichloromethane. The extract was washed with water (3 × 20 mL), dried with anhydrous Na_2_SO_4_ and vacuum evaporated. Pure compounds were obtained by CC.

4-allyl-2-amino-6-methoxyphenol (**8**): Compound **8** was synthesized from the nitroeugenol derivative **7** by reduction of the nitro group. Pure compound **8** was obtained by CC (1:8–1:7, AcOEt in hexane), as white crystal (94.5 mg, 55% yield); melting point: 107–108°C. IR (film) υ_max_/cm^−1^: 3308 (OH), 3372 (NH), 3308 (C=CH Ar), 3072 (CH=CH_2_), 1608 (C=C), 1221 (C-N), 1190 (C-O), 1132 (C-O), 1080, 896. ^1^H NMR: δ 3.24 (2H, d, *J* = 6.7 Hz, H-1′); 3.69 (2H, b. s, NH_2_); 3.84 (3H, s, OCH_3_); 5.05 (2H, m, H-3′); 5.33 (1H, s, OH); 5.93 (1H, m, H-2′); 6.19 (1H, s, H-3); 6.25 (1H, s, H-5). ^13^C NMR: δ 40.1(C-1′); 55.9 (OCH_3_); 101.8 (C-5); 109.4 (C-3′); 115.3 (C-3); 131.1 (C-4); 131.6 (C-6); 138.0 (C-1); 134.0 (C-2′); 137.9 (C-1); 146.6 (C-2).

2-acetamide-4-allyl-6-methoxyphenyl acetate (**9**): Compound **9** was synthesized by acetylation of **8**. Pure compound **9** was obtained by CC (1:9–1:5, ethyl acetate in hexane) as white crystal (103 mg, 70%); melting point: 146–149 °C; IR (film) υ max/cm^−1^: 3320 (N-H), 3060-2800 (C=C Ar), 1750 (C=O), 1250 (C-CO-O), 1120 (C-O), 900-730 (C-H Ar). ^1^H NMR: δ 2.15 (3H, s, C-CH_3_); 2.35 (3H, s, C-CH_3_); 3.36 (2H, d, *J* = 6.5, H-1′); 3.80 (3H, s, OCH_3_); 5.10 (2H, m, H-3′); 5.95 (1H, m, H-2′); 6.56 (1H, s, H-3); 7.14 (1H, b.s, -NH); 7.70 (1H, s, H-5). ^13^C NMR: δ 20.5 (CH_3_CO_2_); 24.7 (CH_3_CO_2_); 40.4 (C-1′); 56.0 (OCH_3_); 108.0 (C-5); 113.9 (C-3′); 116.3 (C-3); 127.9 (C-4); 130.9 (C-6); 136.8 (C-2′); 138.9 (C-1); 150.7 (C-2); 168.0 (CH_3_CO_2_); 168.4 (CH_3_CO_2_).

4-allyl-2-methoxy-6-nitrophenyl acetate (**10**): Compound **10** was synthesized by acetylation of **7**. Pure compound **10** was obtained by CC (1:9–1:7, ethyl acetate in hexane) as yellow crystal (0.45 g, 75%); melting point: 60–61 °C; IR (film) υ max/cm^−1^: 3100-2800 (C=C Ar), 1750 (C=O), 1550 (−NO_2_), 1250 (C-CO-O), 1250-1050 (C-O), 900–730 (C-H Ar). ^1^H NMR: δ 2.36 (3H, s, CO-CH_3_); 3.43 (2H, d, *J* = 6.5 Hz. H-1′); 3.87 (3H, s, -O-CH_3_); 5.17 (2H, m, H-3′); 5.92 (1H, m, H-2′); 7.03 (1H, s, H-3); 7.43 (1H, s, H-5). ^13^C NMR: δ 20.2 (CH_3_CO_2_); 39.5 (C-1′); 56.5 (OCH_3_); 116.0 (C-5); 117.1 (C-3′); 117.4 (C-3); 132.1 (C-4); 135.2 (C-6); 139.1 (C-2′); 142.4 (C-1); 152.6 (C-2); 167.9 (CH_3_CO_2_).

5-allyl-1,2-dimethoxy-3-nitrobenzene (**11**): Potassium carbonate (1.0 g, 7.25 × 10^−3^ mol) was added to a stirred solution of 7 (600 mg, 2.87 × 10^−3^ mol) in dry acetone (60 mL). Then, dimethyl sulphate (1.2 mL, 1.27 × 10^−2^ mol) was added and the reaction was left to continue over night under reflux. Complete disappearance of the starting product was confirmed by TLC (1:3 ethyl acetate: hexane). The crude reaction product is diluted in acetone (30 mL), water (50 mL), and extracted with dichloromethane (3 × 50 mL). The extract is dried with anhydrous Na_2_SO_4_ and vacuum evaporated. Pure product (513 mg, 80%) was obtained by CC (1:9–1:7, ethyl acetate in hexane as marron oil. Compound **11**: IR (film) υ max/cm^−1^: 3100-2800 (C=C Ar), 1550 (-NO_2_), 1400-1000 (C-O), 1000-650 (H_2_C=CH_2_). ^1^H RMN: δ 3.38 (2H, d, *J* = 6.4 Hz, H-1′); 3.90 (3H, s, OCH_3_); 3.95 (3H, s, OCH_3_); 5.14 (2H, m, H-3′); 5.91 (1H, m, H-2′); 6.92 (1H, s, H-3); 7.16 (1H, s, H-5). ^13^C RMN: δ 39.6 (C-1′); 56.4 (OCH_3_); 61.9 (OCH_3_); 115.6 (C-6); 116.3 (C-3′); 117.3 (C-3); 135.7 (C-4); 136.3 (C-2′); 141.1 (C-5); 144.7 (C-1); 153.9 (C-2).

5-allyl-2,3-dimethoxyaniline (**12**): Aniline **12** was synthesized from the nitroeugenol derivative **11** by reduction of the nitro group. Pure compound **12** was obtained by CC (1:8–1:5, AcOEt in hexane) as marron oil (45.7 mg, 46% yield). Compound **12**: IR (film) ʋ_max_/cm^−1^: 3500-3300 (NH_2_), 3100-2800 (C=C Ar), 1400-1000 (C-O), 1000-650 (H_2_C=CH_2_). ^1^H NMR: δ 3.25 (2H, d, *J* = 6.4 Hz, H1′); 3.69 (2H, b.s, NH_2_); 3.79 (3H, s, OCH_3_); 3.82 (3H, s, OCH_3_); 5.07 (2H, m, H-3′); 5.95 (1H, m, H-2′); 6.17 (1H, s, H-3); 6.23 (1H, s, H-5). ^13^C δ NMR: 40.3 (C-1′); 55.6 (OCH_3_); 59.9 (OCH_3_); 102.7 (C-6); 108.8 (C-3′); 115.7 (C-3); 134.3 (C-4); 136.2 (C-2′); 137.5 (C-5); 140.2 (C-1); 152.8 (C-2).

N-(5-allyl-2,3-dimethoxyphenyl) acetamide (**13**): Compound **13** was synthesized by acetylation of **12**. Pure compound **13** was obtained by CC (1:5-1:3, ethyl acetate in hexane) as white crystal (64 mg, 53%); melting point: 78–79 °C; IR (film) υ_max_/cm^−1^: 3300 (N-H), 3100-2800 (C=C Ar), 1715 (R-(CO)-R’), 1400-1000 (C-O), 1000-650 (H_2_C=CH_2_). ^1^H NMR: δ 2.18 (3H, s, CH_3_CON); 3.31 (2H, d, *J* = 6.4 Hz, H-1′); 3.82 (3H, s, OCH_3_); 3.82 (3H, s, OCH_3_); 5.07 (2H, m, H-3′); 5.92 (1H, m, H-2′); 6.47 (1H, s, H-3); 7.81 (1H, s, H-5); 7.83 (-HN). ^13^C NMR: δ 24.7 (CH_3_CON); 40.3 (C-1′); 55.6 (OCH_3_); 60.5 (OCH_3_); 107.6 (C-6); 112.4 (C-3′); 115.7 (C-3); 131.7 (C-4); 135.4 (C-2′); 136.2 (C-5); 137.1 (C-1); 151.7 (C-2); 168.2 (CH_3_CON).

### 4.3. Biological Assays

#### 4.3.1. B. cinerea Isolates

PN2 is a native resistant isolate of *B. cinerea* obtained from cherry. It was maintained and grown under conditions previously described [33]. Its sensibility to current fungicides was tested following a described method [34]. Briefly, to test the single fungicide resistance spectra, of collected *Botrytis* isolates, discriminatory fungicide concentrations of the following fungicides were used: carbendazim 5 µg/mL; cyprodinil 5 µg/mL; fenhexamid 5 µg/mL; iprodion 5 µg/mL; tebuconazole 7 µg/mL; pyrimethanil 10 µg/mL; boscalid 5 µg/mL and tolnaftate 5 µg/mL. These concentrations were chosen according to IC_50_ values of single fungicide resistant strains [35]. In addition, the virulence of PN2 was evaluated by evaluating the damage produced by *Botrytis* isolate on a tomato. Tomatoes fruit were inoculated with 10 µL of conidia suspension (2.5 × 10^6^ conidia/mL) and incubated at 22 °C. The photographical recording was made after 1–4 days post-inoculation to evaluate fruit damage. The Gamborg medium and 7AC isolate (gently provided by Diagnofruit S.A. Santiago, Chile) were used as negative and positive controls, respectively.

#### 4.3.2. Effect of Eugenol Derivatives on Mycelial Growth of *B. cinerea* in Solid Media

Fungitoxicity of eugenol and synthetic derivatives, and of commercial fungicide BC-1000, was assessed using the radial growth test on malt-yeast extract agar [36]. All the compounds, except BC-1000, were applied in dichloromethane solution at different final concentrations (20, 40, 80, 160 ppm). Eugenol and its synthetic derivatives were completely soluble at all tested concentrations. An aliquot of these solutions (200 µL) was added to 7 mL of malt-yeast extract agar. The amount of added dichloromethane was identical in controls and treatment assays. The medium with presence or absence of synthetized compounds as well as BC-1000 was poured into 6 cm diameter Petri dishes. Dishes were left open in a biosecurity hood for 40 min to remove solvent. After solvent evaporation, Petri dishes were inoculated with 5 mm diameter agar discs with thin mycelium of *B. cinerea*. Cultures were incubated in the dark at 22 °C during several days. Mycelial growth diameters were measured daily, and inhibition percentages were calculated using the following equation
(1)$$\mathrm{Growth}\;\mathrm{inhibition}\;(\%)=\frac{\left(d_{C}-d_{0}\right)-\left(d_{S}-d_{0}\right)}{\left(d_{C}-d_{0}\right)}\times 100$$
where *d_C_*, *d*_0_ and *d_S_* represent diameters (in mm) of blank control fungus, fungus agar disc, and compound-treated fungus. Inhibition percentages were plotted as a function of fungicide concentration using a dose-response equation, for all tested compounds. The IC_50_ values were obtained by fitting of data to this curve. Plotting of the data, fitting, and IC_50_ calculation were carried out with Origin v8.0 (OriginLab, Northhampton, MA, USA). Significant differences were evaluated with a two-way analysis of variance (Tukey’s test; *p* < 0.05).

#### 4.3.3. Effect of Eugenol and its Derivatives on ROS Production by *Botrytis cinerea*

The production of reactive oxygen species was measured following a described procedure [36,37]. Menadione was used as a positive control. This molecule has been described as an inducer of ROS generation, since it is capable of both redox cycling and arylating nucleophilic substrates by Michael addition [37,38]. The ROS was evaluated using ROS-GLOtm H_2_O_2_ assay kit (Promega, Madison, WI, USA) [37]. Briefly, conidia (79 µL) were plated in a 96-well plate at a concentration of 1 × 10^5^ conidiam L^−1^/well. Later, each well was incubated in presence of each compound at their IC_50_ concentration, 20 µL of buffer substrate H_2_O_2_ and cultured for 2 h at 21 ± 1 °C. After this period of incubation, 100 µL of ROS-GLOtm reagent was added to each well and incubated for 20 min at room temperature. The ROS production was measured using a luminometer (Tecan infinite m200pro; Tecan Group Ltd., Hombrechtikon, Switzerland). Mean values with at least significant difference (*p* < 0.05) were considered.

## 5. Conclusions

A series of eugenol derivatives has been synthesized and their antifungal activity on mycelial growth of *B. cinerea* has been evaluated. The growth inhibition activity depends on the chemical structure of eugenol derivatives. An analysis of the structure-activity relationship suggests that these compounds act on fungus by two mechanisms, i.e., accumulation on the fungus membrane and chemical reactions with unsaturated chains or enzymatic-mediated reduction. The first is controlled by the lipophilic character of these molecules and the second is due to the presence of strong electron withdrawing groups in the aromatic ring. This action leads to the disruption of fungus membranes and ROS production. Our results show that some eugenol derivatives have a much higher antifungal activity or, at least, comparable to the commercial fungicide BC-1000. Finally, the high activity values presented by the nitro compounds make them potential antifungal agents for the chemical control of *B. cinerea*.

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
