# Peer review of "Antifungal Activity of Eugenol Derivatives against Botrytis Cinerea"

_molecules, 2019, doi:10.3390/molecules24071239_

Round 1
Reviewer 1 Report
The manuscript “Antifungal activity of eugenol derivatives against Botrytis cinErea” presented by Carrasco and collaborators is a very interesting work. I recommend this paper for publication in MOLECULES after minor revision.
Some minor details:
1. I consider that IC50 is more appropriate than EC50 when antifungal activity is evaluated.
2. Abstract: Instead of “effective natural control” it should appear “effective control with inspired natural compounds”
3. Line 102: Compound 7 instead of compound 8; and compound 8 instead compound 9.
4. Table 1: What does it mean “I” for compound 8 and “D” for compound 12? These details should appear as footnote of the table.
5. Caption Figure 6: Compounds 7 and 14 should be in bold. It is necessary to check all the manuscript (for example, see line 313), as this is a common mistake.
6. Scheme 2: The arrows are misplaced. The connection of compound 7 with 10 must be direct. The arrow from 7 to 10 and from 7 to 11, must be independent.
7. Caption Scheme 2: In d) “potassium carbonate” is missing.
8. Line 288 and following “-1” must be superscript for “cm-1”
9. Line 290 and all over the manuscript (OCH3); and line 308 and others (-O-CH3). It is necessary to homogenize this formula. I would suggest “OCH3”.
10. Line 299 and following: It is not necessary to add “H-1’ [-CH2-]”. Just “H-1´” is enough.

Author Response
Some minor details:
1. I consider that IC50 is more appropriate than EC50 when antifungal activity is evaluated.
We have changed EC50 by IC50 in the whole manuscript
2. Abstract: Instead of “effective natural control” it should appear “effective control with inspired natural compounds”
This phrase has been rewritten
3. Line 102: Compound 7 instead of compound 8; and compound 8 instead compound 9.
The numbers have been corrected
4. Table 1: What does it mean “I” for compound 8 and “D” for compound 12? These details should appear as footnote of the table.
A footnote has been added to Table 1
5. Caption Figure 6: Compounds 7 and 14 should be in bold. It is necessary to check all the manuscript (for example, see line 313), as this is a common mistake.
We have made this change and checked all the manuscript
6. Scheme 2: The arrows are misplaced. The connection of compound 7 with 10 must be direct. The arrow from 7 to 10 and from 7 to 11, must be independent.
The scheme has been corrected
7. Caption Scheme 2: In d) “potassium carbonate” is missing.
The figure caption has been completed
8. Line 288 and following “-1” must be superscript for “cm-1”
This typo has been corrected in all manuscript
9. Line 290 and all over the manuscript (OCH3); and line 308 and others (-O-CH3). It is necessary to homogenize this formula. I would suggest “OCH3”.
This suggestion has been adopted
10. Line 299 and following: It is not necessary to add “H-1’ [-CH2-]”. Just “H-1´” is enough.
This change has been made
Reviewer 2 Report
The Authors report the ANTIFUNGAL ACTIVITY OF EUGENOL 2 DERIVATIVES AGAINST BOTRYTIS CINEREA but just one isolate was used in the assay. Considering the high variability characterizing Botrytis cinerea evaluations must be increased to isolates representative of the variability of the wild type populations. Reading is difficult also because the order of scheme and tables is not appropriate. Discussion includes results so I suggest to better redefine the order of the presentation of results and discussion.
Author Response
The Authors report the ANTIFUNGAL ACTIVITY OF EUGENOL 2 DERIVATIVES AGAINST BOTRYTIS CINEREA but just one isolate was used in the assay. Considering the high variability characterizing Botrytis cinerea evaluations must be increased to isolates representative of the variability of the wild type populations. Reading is difficult also because the order of scheme and tables is not appropriate. Discussion includes results so I suggest to better redefine the order of the presentation of results and discussion.
The variability of Botrytis cinereal is a real problem and we believe that this is a very interesting point to deal with. However, the main goal of this work is to evaluate the inhibition activities of a family of structural-related chemical compounds. This can be accomplished using just one isolate of the fungus. We agree with this reviewer that it could be quite interesting to study the effect of fungus variability but clearly this is beyond the aim of this work.
Reviewer 3 Report
The manuscript is about a series of phenylpropanoids derived from eugenol synthesized and characterized to be used to control Botrytis cinerea, agent of gray mold in several fruit crops. The MS has been well prepared and the results are quite interesting. I recommend that the MS is accepted for publication in Molecules after minor revision.

Author Response
Minor revisions suggested by this reviewer were attached to the manuscript. These are the following:
Title. Botrytis cinerea should be in italics.
OK.
Line 77. Reference 26 is in italics. Change
Done
Figure 3. Change figure caption
Now it reads “Virulence assay of damage produced by Botrytis on tomato after inoculating fruit with 10 μL of conidia suspension (2.5 × 106 conidia / mL) and incubated at 22 °C. Gamborg medium was used as negative control, whereas 7AC was used as positive control. Acronyms: B05.10 corresponds to a model isolate; PN2, corresponds to a multiresistant native isolate; 7AC, corresponds to a native isolate resistant to fenhexamid (gift gently provided by Diagnofruit S.A.); dpi means days post inoculation”
Reviewer 4 Report
Reviewer #: The manuscript entitled “Antifungal activity of Eugenol derivatives against Botrytis cinerea” provided by Andrés F. Olea et al. showed the potential application of a series of phenylpropanoids derived from eugenol as antifungal compounds.
The authors claimed that the antifungal activity of phenylpropanoid compounds is higher than that reported for the commercial fungicide BC1000.
In my view, this paper could be improved by addressing minor issues prior to publication.
Minor comments:
Q1. Please provide additional explanation of a correlation with the antifungal activity and ROS production in “Results” or “Discussion” section.
Q2. Page 4, Figure 4. Please change Figure 4 with higher resolution than original data, if it is possible.
Q3. I recommend proofreading of the original manuscript by a native speaker.
Author Response
The manuscript entitled “Antifungal activity of Eugenol derivatives against Botrytis cinerea” provided by Andrés F. Olea et al. showed the potential application of a series of phenylpropanoids derived from eugenol as antifungal compounds.
The authors claimed that the antifungal activity of phenylpropanoid compounds is higher than that reported for the commercial fungicide BC1000.
In my view, this paper could be improved by addressing minor issues prior to publication.
Minor comments:
Q1. Please provide additional explanation of a correlation with the antifungal activity and ROS production in “Results” or “Discussion” section.
This is a good point and we have added the following phrase to establish the existing correlation between ROS production and inhibition activity. “Interestingly, the magnitude of ROS production follows the same order of IC50 values, i.e. 15 > 11 > 7. In other words, formation of ROS correlates quite well with the inhibition activity suggesting that nitro compounds are acting on fungus membrane by generating reactive oxygen species and by alteration of membrane in similar way to that reported for eugenol both in fungi and bacteria”
Q2. Page 4, Figure 4. Please change Figure 4 with higher resolution than original data, if it is possible.
We have changed Figure 4.
Q3. I recommend proofreading of the original manuscript by a native speaker.
A complete proofreading has been made to the manuscript.
Round 2
Reviewer 2 Report
Sorry but according to the title reporting "ANTIFUNGAL ACTIVITY OF EUGENOL 2 DERIVATIVES AGAINST BOTRYTIS CINEREA" I do not believe the obtained results using a single strain can support the antifungal activity of Eugenol derivates against Botrytis cinerea. So please at least change the title in "POTENTIAL ANTIFUNGAL ACTIVITY OF EUGENOL 2 DERIVATIVES AGAINST BOTRYTIS CINEREA"